# Progress in the Application of Portable Ultrasound Combined with Artificial Intelligence in Pre-Hospital Emergency and Disaster Sites

**DOI:** 10.3390/diagnostics13213388

**Published:** 2023-11-06

**Authors:** Xing Gao, Qi Lv, Shike Hou

**Affiliations:** 1Tianjin University Tianjin Hospital, Tianjin 300211, China; gaoxingjiayy@163.com; 2Institution of Disaster and Emergency Medicine, Tianjin University, Tianjin 300072, China; 3Key Laboratory of Medical Rescue Key Technology and Equipment, Ministry of Emergency Management, Tianjin 300072, China

**Keywords:** portable ultrasound system, first aid of trauma, artificial intelligence

## Abstract

With the miniaturization of ultrasound and the development of artificial intelligence, its application in disaster scenes and pre-hospital emergency care has become more and more common. This study summarizes the literature on portable ultrasound in pre-hospital emergency and disaster scene treatment in the past decade and reviews the development and application of portable ultrasound. Portable ultrasound diagnostic equipment can be used to diagnose abdominal bleeding, limb fracture, hemopneumothorax, pericardial effusion, etc., based on which trauma can be diagnosed pre-hospital and provide guiding suggestions for the next triage and rescue; in early rescue, portable ultrasound can guide emergency operations, such as tracheal intubation, pericardial cavity puncture, and thoracic and abdominal puncture as well as improve the accuracy and timeliness of operation techniques. In addition, with the development of artificial intelligence (AI), AI-assisted diagnosis can improve the diagnosis level of ultrasound at disaster sites. The portable ultrasound diagnosis system equipped with an AI robotic arm can maximize the pre-screening classification and fast and concise diagnosis and treatment of batch casualties, thus providing a reliable basis for batch casualty classification and evacuation at disaster accident sites.

## 1. Introduction

At present, under the background that medical equipment, medicine and biology technologies, 5G network and artificial intelligence have entered into deeply integrated development, the demand for AI enhanced medical technology is growing stronger in the development of the emergency trauma medicine field so as to protect the lives of the injured using the optimal resources. The development of AI has gone through three stages (as shown in Figure 1). Ultrasound Doppler is a technique for studying the Doppler effect produced by the reflection or scattering of ultrasound waves by a moving object. It is widely used in clinical practice for the diagnosis of the heart, blood vessels, blood flow, and fetal heart rate [1,2,3,4]. At present, miniature ultrasound is a trend of development—the palm-type portable ultrasonic diagnosis system is a medical ultrasonic device which is similar to the size of a mobile phone and is convenient to control by combining a host computer and a probe (as shown in Figure 2). It has the advantages of multi-platform and multi-terminal Wi-Fi connection, real-time clear ultrasound images, flexible and portable ergonomic design, seamless switching of system interface, the net weight of the main unit being only 260 g, etc. [5,6]. Images obtained using ultraportable handheld ultrasound (HHU) are comparable to those obtained with traditional machines but create unique issues regarding billing and data management. The potential benefits of a handheld advanced imaging system are undeniable. With further technological developments, the gap in functionality between handheld and cart-based systems will continue to decline. The comparison between the two is detailed in (Table 1).

With the development of microelectronics technology, its use in medical pre-screening and triage is becoming more and more widespread, especially in the field of pre-hospital emergency. Currently, with the advent of the era of artificial intelligence, the research concept of Vilchis et al. in the robot remote ultrasonic inspection system [8]—combined with 5G’s major breakthroughs and rapid progress in wireless air interface technology, load-bearing network transmission technology, key technologies of core networks, and other aspects—have fully participated in the network structure and business model construction of future remote and mobile ultrasonic in 5G-supported “Internet transmission + edge cloud application + core cloud storage” [9]. Moreover, 5G technology redefines remote and mobile ultrasound, providing remote ultrasound services and extending medical services to outdoor and even more complex field scenarios [10].

AI is a description of the operation mode of an intelligent machine (computer) which imitates human intellectual behavior [11]. Deep learning methods are an important technology for AI development. Convolutional neural networks (CNN) are one of the most popular deep learning architectures and have made great progress in various tasks such as image classification, target detection, and target segmentation [12,13]. The AI-robotic-arm-assisted ultrasound imaging system can realize remote imaging under the control of a physician (as shown in Figure 3). Portable ultrasound diagnostic equipment can be used in an emergency; anesthesia and critical care departments with an AI robotic arm—which is convenient, immediate, and can be remotely guided—are increasingly valued by physicians. The combination of remote physicians and on-site AI robotic arm systems can work together in a shared manner, making intelligent treatment and remote detection possible, especially since portable ultrasound diagnosis systems equipped with AI robotic arms have unique advantages. This combination can overcome the shortcomings of ultrasound through remote operation, co-assist, or even an autonomous system. It can complete multi-terminal interaction between ultrasound instruments and equipment, instruments and casualties, and doctors and physicians at the emergency scene, and simultaneously realize real-time and safe upload and playback of dynamic ultrasound images, thus achieving convenient deployment of ultrasound AI business and forming a cloud platform for ultrasound image big data mining. It thereby redefines ultrasound instruments and equipment and upgrades them from mere medium inspection instruments to intelligent diagnostic equipment. Multi-scene compatibility leads to increased demand for storage space. Nevertheless, the current palm-portable ultrasound diagnostic system is equipped with spare probes, a memorizer, power adapters, iodine disinfectant, puncture needles, and other items in complex scenarios, which—combined with its own advantages of light weight, small size, and flexible mobility—can be better applied in pre-hospital emergency, disaster scene and battlefield environment and also solves the need of storage space [14,15]. During the global health crisis period in 2020, ultrasound radiologists combined with remote portable ultrasound diagnostic systems equipped with AI robotic arms to create a more reliable platform for the treatment of novel coronavirus patients, reducing physician–patient contact and unnecessary consumption of medical staff according to the specific circumstances of the time, playing an important role in Lei Shen Shan hospital and some general hospitals that received patients suffering from novel coronavirus combined with other diseases [16].

**Figure 2 diagnostics-13-03388-f002:**
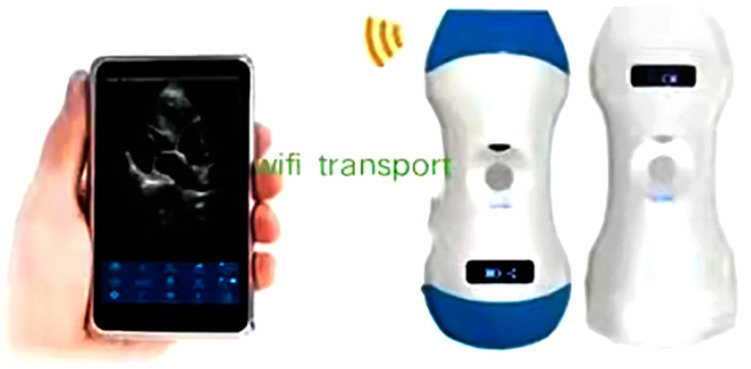
Portable ultrasound [5,15].

In the field of emergency medicine and pre-hospital examinations, the AI robotic arm of portable ultrasound diagnostic system uses wireless network as a transmission link to display instant images on cell phones or iPad, and with the help of 5G technology, the detected information is recognized in time and provided to doctors as a diagnostic reference [17]. As a kind of visualized precision medical instrument, in the field of trauma treatment, the portable ultrasound diagnostic system equipped with AI robotic arm can improve the efficiency of on-site medical personnel. Especially in the field of trauma emergency, the disaster accident site and pre-hospital emergency casualty treatment have the characteristics of harsh rescue environment, high density of casualties, complex injuries, etc. [18]. The palm-type portable ultrasound diagnostic system equipped with an AI robotic arm can maximize the pre-screening and classification of batch casualties and fast and concise diagnosis and treatment, thus minimizing diagnostic errors and complications, avoiding major medical accidents and medical disputes, and providing a reliable basis for batch casualty classification and evacuation at the scene of disaster accidents [6,19,20,21,22]. At present, the United States has started to set up relevant departments to look forward to the future development of AI and its broad impact on society [23].

## 2. Application of Portable Ultrasound in Pre-Hospital Trauma Emergency

The scene of future wars and major emergencies, large numbers of casualties will be produced in a short period of time, the complex types of injuries include multiple injuries, compound injuries, and special injuries (see Table 2), all kinds of critical injuries are often accompanied by cardiac and respiratory arrest; hemorrhagic shock; damage to cardiovascular, lung, spine, and abdominal organs. If the on-site treatment is not timely, the mortality rate of the sick and injured will greatly increase. “Mass casualty incident” (MCI) is defined as having two or more patients injured in such a way that the required medical response exceeds the resources available to care for the victims. Patient survival during an MCI depends on the ability of responders to rapidly and accurately triage patients to the appropriate level of care. Most currently applied triage schemes rely on history and physical examination, both of which can be unreliable and/or difficult to obtain in MCI settings [24,25]. Use of an adjunct that facilitates this triage process may be the key to potentially improving care, and thus survival, during MCIs [26]. There is a consensus among experts who support ultrasound screening of mass casualties as a quick and effective means of detecting torso and internal injuries for diagnosis, triage and simple treatment [27,28,29,30,31,32,33]. Those with high cumulative points in the trauma index classification will be screened and evaluated first. The next step of examination will be carried out actively, and then the patients will be quickly transported to the next stage of treatment. According to the trauma scoring system, injury classification of the entire body can be completed within 5 min by using portable ultrasound [34], which provides more reliable and convenient diagnosis and treatment basis for serious injury control and evacuation. Portable ultrasound was used in the triage of patients after the 2010 Haitian earthquake. The investigators noted that the results of their ultrasound examination influenced care in 70% of cases [35]. Ultrasound use following the Wenchuan earthquake disaster in 2008 reported a sensitivity of 91.9% and a specificity of 96.6% for the diagnosis of abdominal injuries [36]. In general, the order of ultrasound examination is shown in Figure 4.

### 2.1. Application of Portable Intelligent Ultrasound in the Diagnosis of Chest Trauma

Portable ultrasound diagnostic systems can be used in the field of trauma emergency chest emergency examination to determine the cause of the patient’s dyspnea, such as the presence of pulmonary edema, pneumonia, pleural effusion, etc.; have similar diagnostic effects to chest CT; and are better than chest X-rays [37,38]. The portable ultrasound diagnostic system is especially suitable for rapid classification and diagnosis of closed lung injuries in a large area within a short period of time under major trauma conditions [39] and can quickly identify life-threatening injuries such as pericardial tamponade, pneumothorax, hemothorax, etc. (as shown in Figure 5), without radiological damage and can perform dynamic and repeated examinations as well as assist physicians in determining whether other examinations are needed. In pre-hospital emergencies, emergency physicians can use portable ultrasound diagnostic systems to identify fluid in the patient’s body cavity and perform rapid assessment and diagnosis. Based on previous experience, the focused assessment with sonography for trauma (FAST) method, which uses a portable ultrasound system with an AI arm to detect intra-abdominal bleeding, has become the preferred technique for emergency personnel to assess abdominal injuries [40].

In the evaluation of thoracic trauma, the examination includes subcostal views and the cardiac region, and if a pericardial effusion is found, pericardial tamponade treatment along with puncture decompression is required. Ultrasound examination on the anterior aspect of the chest is useful in identifying hemothorax and pneumothorax with higher specificity and sensitivity compared to X-ray examination [41]. In addition, the portable ultrasound system with AI is also suitable for penetrating trauma and blunt trauma assessment of the chest, which is more suitable for patients with unstable vital signs [42]. In cases of respiratory cardiac arrest, cardiopulmonary resuscitation (CPR) is required immediately. The role of portable ultrasound diagnostic system in CPR is becoming more and more important, and it is easier and faster to move around to assist doctors in determining the cause of the disease, such as massive cardiac infarction, hypovolemia, or pulmonary embolism. Additionally, it can also identify heart contractions in the absence of a pulse, assess CPR’s effects, evaluate patient prognosis in order to prepare for the next step of emergency treatment. Echocardiography can be used to assess cardiac function, check for the presence of pericardial effusion, diagnose cardiac conditions quickly and accurately, and reduce casualties in emergency situations with ultrasound-guided pericardial effusion puncture and catheter drainage by robotic arm [43]. It can also be used for airway assessment, which can help to confirm the position of tracheal intubation to ensure a smooth airway, and for post-cardiopulmonary resuscitation to assess the patient’s myocardial function and internal blood volume status to provide a basis for the appropriate treatment modality.

AI also can use the modular software system to automatically assist the analysis of cardiovascular ultrasound images, simplify the vascular ultrasound examination process, reduce the dependence on the operator, and shorten the examination time [45].

### 2.2. Application of Portable Intelligent Ultrasound in the Diagnosis of Abdominal Trauma

Focused assessment sonograph for trauma (FAST) is a medically proven method for the initial evaluation of closed intra-abdominal injuries. The portable ultrasound diagnostic system takes less time than traditional ultrasound and CT and is more mobile than CT and conventional ultrasound in terms of accuracy and sensitivity in exploring the abdominal cavity, with no significant differences [46]. It can be repeatedly and continuously examined, and the change in the amount of fluid and blood can be examined in a dynamic situation, which facilitates the examination and medical judgment of the disease [47].

In addition, portable ultrasound diagnostic system has the advantages of small size, light weight, and ease of transport [48]. Pre-hospital use of the FAST robotic arm equipped with a portable ultrasound instrument can complete a certain examination work, and with no impact on the body, it can be used multiple times. It also has the function of completing intra-abdominal, vascular, and uterine cavity fluid examinations and has superior penetration and physical clarity. It has the highest accuracy in examining tissue areas where fluid is present and is currently the best examination method, especially for patients with closed abdominal injuries with rapid symptom changes. Recent studies have favored the use of FAST as an aid to casualty classification in the U.S. and European emergency medical systems [49,50]. The results show that portable ultrasound plays a role in rapid injury triage and diagnosis in both pre-hospital emergency systems and airborne helicopter transport systems, especially in disaster accidents and combat rescue. Using portable ultrasound for FAST assessment can improve the utilization efficiency of medical resources and improve the effectiveness of casualty care.

The use of ultrasound in the diagnosis of ectopic pregnancy in pre-hospital emergency can accurately determine the staging of ectopic pregnancy, and the accuracy of Transvagin Scan (TVS) and Transabdominal ultrasound (TAS) diagnosis is higher, which can provide reliable guidance for emergency medical care [51]. In this study, it was noted that FAST can effectively facilitate the management of abdominal wounds, which is of key importance in the process of abdominal trauma treatment [52].

Traumatic pneumoperitoneum is usually caused by a perforated gastrointestinal tract, so prompt examination can assist in the early diagnosis of the patient. Gastrointestinal perforation is usually diagnosed using X-ray or CT. Some investigations have shown that the sensitivity and specificity of abdominal ultrasound for pneumoperitoneum examination are 85–90% and 100%, respectively, and experienced physicians can even detect 1 mL of gas, which is comparable to CT examination [53]. A study indicates from 84 cases of abdominal parenchymal organ damage diagnosed by enhanced CT that 81 were positive according to ultrasonography [54]. In a model simulating the diagnosis and treatment of spleen injury in pigs, the diagnostic accuracy of intelligent ultrasound was higher than that of physicians. This study suggests that combined ultrasound imaging can improve the ability of ultrasound to identify organs, reduce the number of CT tests, improve the speed of detection and analysis of patients with abdominal damage, and increase the success rate of patient resuscitation [55].

In terms of artificial intelligence, Pavlopoulos et al. analyzed liver ultrasound images using a fuzzy neural network. The results show fractal dimension texture analysis (FDTA), spatial gray-level dependence matrix (SGLDM), gray-level co-occurrence matrix (GLCM), gray-level run length statistics (RUNL), and first-order gray-level parameters (FOP). The five feature parameters are trained to the network through the geometric blur device, which can be used to intelligently identify disseminated liver lesions [56].

### 2.3. Application of Portable Intelligent Ultrasound in Musculoskeletal Trauma Diagnosis

In earthquakes, typhoons, tsunamis, and other disaster sites that rescuers cannot swiftly reach in great numbers—and to which large X-rays, CTs, and other equipment is inconvenient to move—the application of portable ultrasound diagnostic system examination can be more convenient to diagnose the patient. Musculoskeletal ultrasound (MSUS) is a new ultrasound examination technology for diagnosing musculoskeletal system diseases through high-frequency ultrasound (3–22 Hz) scans to provide excellent clinical images to clearly show the hierarchical relationship of muscle and other soft tissue and its internal institutions. For example, the application of portable ultrasound with a robotic arm to examine the direct signs of fracture sonograms can determine whether the part is a fracture or a continuity fracture, providing more direct evidence for further treatment. More common after earthquakes is crush syndrome, which is also one of the main causes of death of the injured after earthquakes. Crush syndrome refers to severe damage to the soft tissues of human muscles under the action of blunt external forces, such as crushing and blows, and the main manifestations include swelling, bleeding, local sensory impairment, and even acute renal failure at the damaged area. Portable intelligent ultrasound diagnosis can obtain accurate diagnostic information on suspected lesion sites and at the same time can assist in the diagnosis of crush syndrome by observing the activity level of muscles, tendons, etc. This is the preferred screening method for soft tissue injuries after earthquakes [57].

The portable ultrasonic diagnostic system can detect the degree of soft tissue and bone damage at the trauma site and make timely and accurate injury judgments for patients while preventing the emergence of crush syndrome to a certain extent. It can also judge fluid accumulation in joint cavities and abnormal muscle echogenicity. Through the examination and diagnosis of portable ultrasound, people affected by the earthquake can have an understanding of their own physical condition and also relieve post-earthquake psychological pressure and improve their confidence in their own physical recovery, which also has a positive effect on post-disaster reconstruction.

In addition, with the wide application of high-resolution ultrasound in the skeletal muscle system, sonographers expect to use AI technology to improve the consistency and accuracy of the clinical interpretation of ultrasound imaging of skeletal muscle injury [58].

Real-time guidance by ultrasound can significantly improve the success rate of puncture, especially in facet joint manipulation. For the special complex effusion with separation, ultrasound guidance can avoid the phenomenon of “dry pumping”, which can easily occur in blind wear.

### 2.4. Application of Portable Intelligent Ultrasound in the Diagnosis of Craniocerebral Injury

Craniocerebral injuries account for the second largest proportion of casualties in a disaster setting. Increased intracranial pressure indicates the presence of intracranial hematoma or even the risk of cerebral herniation, which endangers life. Thus, rapid diagnosis and management are critical to patient survival. Portable ultrasound diagnostic systems equipped with AI robotic arms to measure the diameter of optic nerve sheath for intracranial pressure assessment can detect plateau cerebral edema in time. The results recorded using color ultrasound for trauma-induced cerebral edema casualties are quickly conveyed to the rear expert pool through the AI’s memory system, applying the current 5G network, and through feedback from rear expert opinions. Treatment is implemented at the rescue site immediately, providing strong support for the recovery of the function of craniocerebral injury casualties.

At present, artificial intelligence ultrasound combined with 3D technology has been successfully applied in the monitoring of fetal thalamus, choroid plexus, transparent lumen, lateral, fissure and other brain structures and combined with neural network information analysis to make accurate judgments [59].

In addition to the above applications, portable ultrasound combined with AI can also assist doctors in emergency invasive operations, such as tracheal intubation, cricothyroidocentesis, assisted arterial placement, and nerve block anesthesia, which can effectively improve the efficiency and success rate of emergency care.

## 3. Conclusions and Prospects

The application of modern technology combined with artificial intelligence can transform the management of trauma, respiratory distress and cardiac arrest patients, and the images obtained can be comparable to those obtained with conventional ultrasound equipment and save human resources. Ultra-portable handheld ultrasound with augmented reality (AR) technology and remote guidance allows physicians with extensive POCUS experience to remotely guide pre-hospital emergency for personnel in making treatment decisions and ensuring examination qualities. With the continuous improvement of ultrasound technology, diagnosis, and treatment methods and procedures, portable ultrasound diagnostic systems are more advantageous for the diagnosis and management of emergency cases as well as for ultrasound-guided treatment. With the continuous development of technology, Artificial intelligence -assisted portable ultrasound diagnosis systems are showing good prospects for application and are gaining popularity among physicians.

Portable ultrasound intelligent system examination is suitable for patients with unstable or potentially unstable vital signs; can rapidly identify life-threatening trauma, including pneumothorax, hemothorax, and pericardial effusion; can assist clinicians in choosing whether to perform other examinations to help patients receive timely treatment; and can repeatedly and dynamically evaluate trauma patients without radiological damage. In addition, remote operation through 5G technology is also a future development trend [60]. Moreover, as equipment comes closer to true “mass production”, it also becomes more readily accessible and less expensive [61,62].

Currently, an artificial intelligence portable ultrasound inspection system combined with a 5G communication platform helps in battlefield or disaster scene rescue work. Ultrasound robots can perform examination, diagnosis, operation, etc., autonomously—which can be detached from the reliance on manual operation during battlefield rescue—and carry out ultrasound remote examination, image transmission, rapid remote consultation, etc., thus saving human and material resources. Intelligent ultrasound can not only be applied to the diagnosis of organs such as those in the chest and abdomen but also has better diagnostic efficacy for the degree of benign and malignant clinical thyroid, which can effectively improve the clinical specificity and positive value [63]. Additionally, in urological ultrasound, the degree of renal obstruction can be assessed by using quantitative technology with artificial intelligence algorithm, and the sensitivity can reach 100% [64]. As medical technology continues to advance, ultrasound is no longer a blind spot for lung examinations, and intelligent ultrasound can provide important clinical value for lung changes in patients. For the assessment of extravascular lung water in patients with acute respiratory distress syndrome, automated lung ultrasound scoring with artificial intelligence is more sensitive compared to manual groups [3].

With this technology, 5G communication can transmit images and other related data to the rear hospital in real time, and the rear hospital personnel can make a diagnosis, which improves the accuracy of diagnosis to a certain extent. In addition, intelligent robots equipped with ultrasound systems have more advantages—such as resistance to fatigue, ability to adapt to harsh environments, ability to work continuously in low- or zero-oxygen environments, automatic driving and positioning, identification of obstacles, etc.—that are practically significant. However, there are still many problems to be solved in other aspects of artificial-intelligence-equipped portable ultrasound systems, such as its weight and it generally being equipped with only an abdominal probe, which leads to a narrow inspection range. Additionally, robotic-arm-carrying ultrasound systems require professional training. In particular, the probe is prone to instrument failure when touching or falling; otherwise, there are few physicians with relevant expertise in the treatment of injuries at the scene of disaster accidents, and physicians who have not been trained in ultrasound operations have not fully mastered portable ultrasound examination techniques. In addition, machine-stylized diagnosis may create a gap between doctors and patients and weaken the multidisciplinary joint diagnosis and treatment of diseases. Therefore, in order to give full play to the advantages of ultrasound AI in diagnosis, we must properly coordinate between machines and humans.

Currently, the portable ultrasound system equipped with artificial intelligence has characteristics of low power consumption, low latency, high speed, and wide coverage, meanwhile breaking the technical barriers to the development of remote ultrasound robots. Data management and sharing, remote consultation, background analysis, reverse control, and pre-hospital emergency are realized through remote data retrieval [65,66], while the rapidly developing 5G technology achieves real-time synchronous exchange of images, voice, and scenes of the remote ultrasound robot system [67]. Wuhan University Zhongnan Hospital research points out that a portable ultrasound diagnostic system equipped with an AI robotic arm can replace conventional ultrasound to a certain extent to diagnose common diseases and meet daily diagnosis and treatment needs [68]. One of the major drawbacks of conventional ultrasound imaging is the reliance on manual probe positioning and the resulting user dependence as well as the potential for work-related musculoskeletal disorders that can result from this prolonged single-position procedure [69]. Portable diagnostic ultrasound systems can help overcome this problem and facilitate long-term acquisition of image data to observe dynamic processes in the body over time. The combination of portable ultrasonic examination and a 5G communication platform has become an ultrasonic robot that can be deployed to the disaster site, which is worthy of being vigorously promoted. In addition, intelligent robots equipped with ultrasound systems have the advantages of not easily fatiguing, adapting to the darkness and harsh environment of disaster accident sites, and working continuously under conditions such as oxygen deprivation which are incompatible with human beings; at the same time, they have the functions of automatic driving, automatic positioning, automatic identification of road obstacles, etc., and have the practical significance of rescuing and evacuating the injured instead of humans.

The limitations: Although some progress has been made in AI-assisted technology, there is still a lack of systematic multi-center clinical trials and effective validation, and there is still a gap between the requirements of clinical standardization. Ultrasound recognition artificial intelligence algorithm has a certain range of cognitive limitations in the application field of practical ultrasound technology. Although ultrasound artificial intelligence technology effectively helps to improve the recognition and resolution ability of medical ultrasound images, there is still some distance from the requirements of clinical standardization when it is really introduced into clinical practice. Implementation of machine learning and tele-ultrasound applications is likely to alter the point-of-care ultrasound (POCUS) landscape, and physicians should continue to study and embrace these technologies for the improvement of patient care.

In summary, AI is expected to become an effective auxiliary tool for imaging doctors in the diagnostic process in the future. The portable ultrasound system has, to some extent, improved the speed and efficiency of pre-hospital diagnosis and fought for time and human resources for pre-hospital emergency and disaster bulk casualty detection and classification treatment despite carrying intelligent 5G system allowing it to realize remote diagnosis and detection, further improving the accuracy and level of diagnosis and safeguarding the life and health of patients. AI-integrated ultrasound technology has significant advantages in the pre-examination and triage of disaster sites and pre-hospital emergency treatment. Therefore, we forecast that in the future, multi-ultrasound technology and AI development can help greatly in the initial assessment of the sick’s conditions.

## Figures and Tables

**Figure 1 diagnostics-13-03388-f001:**
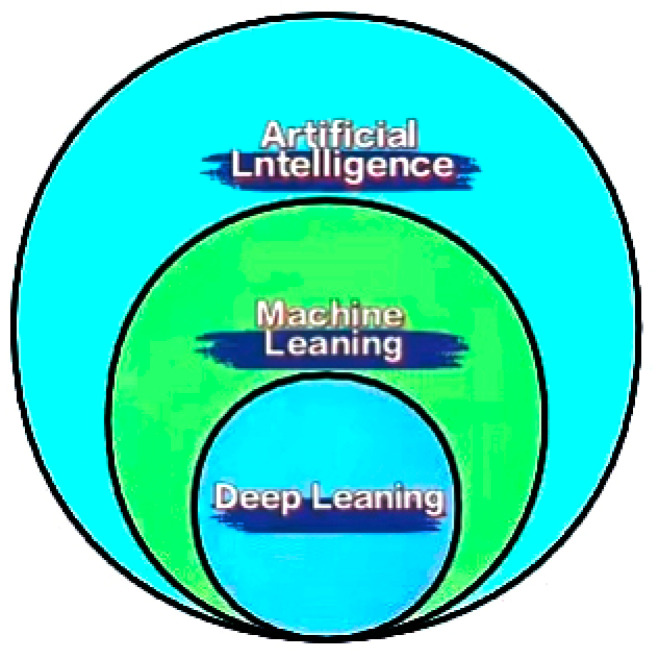
Relationship between machine learning, deep learning and artificial intelligence.

**Figure 3 diagnostics-13-03388-f003:**
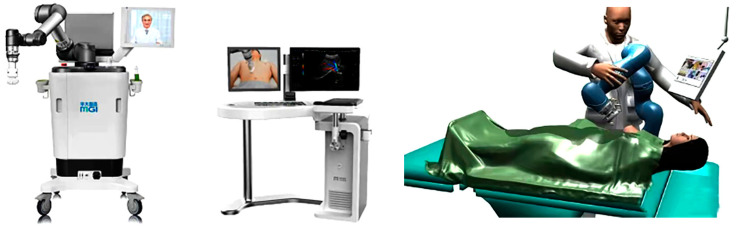
AI robotic arm-assisted ultrasound imaging system and operation flow of wireless color Doppler ultrasound diagnosis system [17].

**Figure 4 diagnostics-13-03388-f004:**
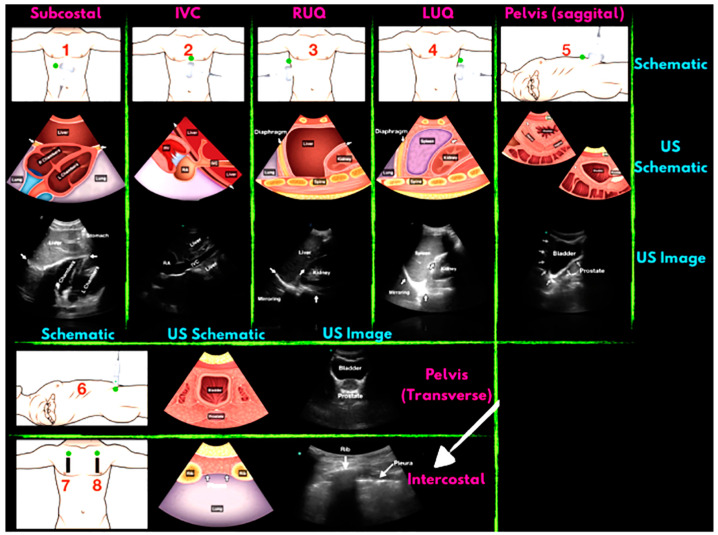
FAST schematic diagram of the examination site and the imaging. Green dots indicate the examination site. IVC, inferior vena cava; RUQ, right upper quadrant; LUQ, left upper quadrant.

**Figure 5 diagnostics-13-03388-f005:**
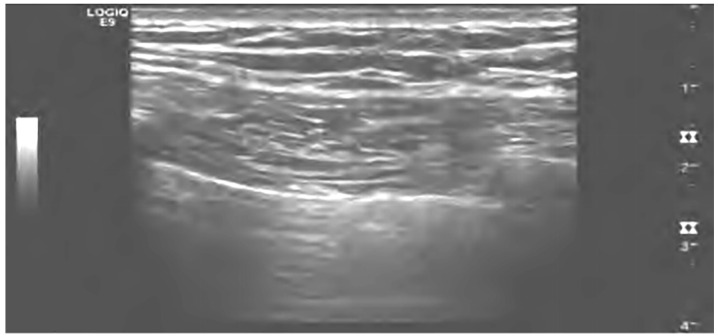
Ultrasound diagnosis of pneumothorax with lung sliding [44].

**Table 1 diagnostics-13-03388-t001:** Portable handheld device and a high-end ultrasound device compare [7].

	Portable Handheld Device	High-End Ultrasound Device
Portability	Easily	Not easy
Prime cost	Low	High
Application scenarios	Primary hospital, trauma emergency treatment scene	Hospital clinic
Image quality	Not good	Preferably
Operability	Ordinary doctor	Ultrasound professional doctor

**Table 2 diagnostics-13-03388-t002:** Injury sites and types.

Injury Sites	Fatal Injury Types
Head and face	Brain stem injury, diffuse cerebral contusion, intracranial hemorrhage
Neck	Carotid artery dissection, high spinal cord injury
Thoracic cavity	Tension pneumothorax, rupture of a large blood vessel in the chest, rupture of the heart, pericardial tamponade
Abdomen	Hepatic and splenic rupture, abdominal aortic rupture, kidney injury
Pelvic cavity	Pelvic fracture with rupture of the common iliac artery
Arms and legs	Amputation bleeding, crush syndrome

## Data Availability

Data sharing is not applicable to this article as no new data were created or analyzed in this study.

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
