# Peer review of "Progress in the Application of Portable Ultrasound Combined with Artificial Intelligence in Pre-Hospital Emergency and Disaster Sites"

_diagnostics, 2023, doi:10.3390/diagnostics13213388_

Round 1

Reviewer 1 Report

Comments and Suggestions for Authors

No criticisms from this reviewer for this monumentally important display of the clinical benefit of current technologic application.

Author Response

Thank you professor

Reviewer 2 Report

Comments and Suggestions for Authors

Dear authors,

I received your submission as a reviewer and found that you have tried to highlight the importance and usefulness of AI+US in prehospital setting. In my viewpoint, the topic is attractive, however, the text needs revision.

First of all, I think that considering better continuity would be so important throughout the whole text.

Secondly, the usefulness of US in pre-hospital setting is highly under debate, as so many believe that except in cases of tension pneumothorax, it could not alter the management process of trauma patients, and I invite you to point to such an important controversies regarding the use of US in pre-hospital setting. Check the below references in this regard:

https://doi.org/10.1016/j.injury.2015.07.007

https://link.springer.com/article/10.1007/s00068-023-02226-8

Finally, when it comes to conclusion in review articles, it is better to just make suggestion. I recommend you to revise your text appropriately.

Kind regards,

Author Response

General comment: The authors address a very interesting topic. The importance and usefulness of AI+US in prehospital setting.

Response: Thank you very much for acknowledging the importance and usefulness of this topic.

Point 1:  You think that considering better continuity would be so important throughout the whole text.

Response 1: Thank you for your careful reading and for recommending literature to us. We have carefully read the literature as well as your suggestions.Our paper is a review study combining artificial intelligence and ultrasound. The design idea of this review is mainly around the advantages of artificial intelligence and ultrasound in the disaster accident site and pre-hospital triage. The continuity of the article has been adjusted.

Point 2: The usefulness of US in pre-hospital setting is highly under debate, as so many believe that except in cases of tension pneumothorax, it could not alter the management process of trauma patients, and I invite you to point to such an important controversies regarding the use of US in pre-hospital setting.

Response 2: Thanks to your suggestion and the literature that you provide,and use it as a reference(lines 114),first of all, the adaptation scenario of this review is the pre-inspection and triage of the disaster accident site and pre-hospital emergency batch injured, so the detection and treatment of the seriously injured is particularly critical.Our primary research objective of this review is the rapid screening procedure, in order to provide rapid treatment for the seriously injured. we have elaborated in the application section(lines 127-133).

Point 3: When it comes to conclusion in review articles, it is better to just make suggestion. I recommend you to revise your text appropriately.

Response 3: Thank you for your suggestions. In the conclusion section, we mentioned the problem,We have corrected the conclusions (lines 354-364).

Reviewer 3 Report

Comments and Suggestions for Authors

This article deals with a topic of extreme interest for the future of emergency medicine, congratulations for the excellent choice.

I would just ask to make some small changes, for example adding the limitations of this study and the sensitivity and specificity for various possible conditions, for example as specified in Table 1, we know that it is not so easy to see on ultrasound a liver, splenic or renal laceration, unless it is massive, so I would highlight how in some situations, such as performing an Echofast for trauma or even for the diagnosis of PNX (If not always mentioned as additional windows of the E-fast), it has good accuracy, as opposed to the already mentioned splanchnic organ trauma where it is not as reliable.

Author Response

General comment:Thank you for the opportunity to review this manuscript.

Given the clinical significance of Portable Ultrasound combined with artificial intelligence and he need of summarizing the recently developed diagnostic tools, this manuscript provides valuable information. 

In order to improve its readability, some technical facts must be taken into account:

Response: Thank you for recognizing the significance of this review, and below are our point-by-point answers to your concerns and the corresponding lines in the current version of the manuscript.

Point: we know that it is not so easy to see on ultrasound a liver, splenic or renal laceration, unless it is massive, so I would highlight how in some situations, such as performing an Echofast for trauma or even for the diagnosis of PNX (If not always mentioned as additional windows of the E-fast), it has good accuracy, as opposed to the already mentioned splanchnic organ trauma where it is not as reliable.

Response : Thank you for your suggestions. First of all, the adaptation scenario of this review is the pre-inspection and triage of the disaster accident site and pre-hospital emergency batch injured, so the detection and treatment of the seriously injured is particularly critical. Ultrasound inspection can provide technical support to the combination of the injured parts of the injured, so ultrasound has obvious advantages over other equipment in the inspection of the batch injuredIn the introduction section.we mentioned the abdominal ultrasound examination, and described it (lines 195-207) and chest ultrasound examination described (lines 127-139).

Round 2

Reviewer 3 Report

Comments and Suggestions for Authors

In line 203 talked about "TVS+TAS" without specifying what these abbreviations mean, table 1 is not clear when it is mentioned in the text, please add this information.

Also add a paragraph where you talk about the limitations of your study.

Regarding your answer, ultrasound inspection we agree that it can be a technical support in prehospital, but as this is a scientific article we need to be more precise. 

How much of an impact can an evaluation performed accompanied by ultrasound have compared with a simple objective examination?

How does a portable ultrasound machine differ in terms of image quality from one present in the emergency department?

Are there studies that have shown that ultrasound use reduces mortality or other end points in the prehospital setting? If not, add it as a limitation.

How much would it cost to equip each operating unit with a portable ultrasound machine? If it is too costly, might the use of portable ultrasound be advisable only for the most severe scenarios?

These are all questions that arise when reading your article, and which you need to answer.

Author Response

Response to Reviewer 3 Comments

General comment: Thank you for the opportunity to review this manuscript.Given the clinical significance of Portable Ultrasound combined with artificial intelligence and he need of summarizing the recently developed diagnostic tools, this manuscript provides valuable information. 

In order to improve its readability, some technical facts must be taken into account:

Response: Thank you for recognizing the significance of this review, and below are our point-by-point answers to your concerns and the corresponding lines in the current version of the manuscript. The continuity of the article has been adjusted.

Point 1:  In line 203 talked about "TVS+TAS" without specifying what these abbreviations mean, table 1 is not clear when it is mentioned in the text, please add this information.

Response: Thank you for your careful reading and suggestion. "TVS+TAS" means Transvagin Scan (TVS) and Transabdominal ultrasound (TAS), which has been explained in the revised MS (line 217).the table1 has been adjusted table2 and explained it in the article(lines 112-117).

Point 2: Also add a paragraph where you talk about the limitations of your study.

Response: Thanks to your suggestion. The limitations of the study have been added in the revised MS (lines 380-391).

Point 3: Regarding your answer, ultrasound inspection we agree that it can be a technical support in prehospital, but as this is a scientific article we need to be more precise. How much of an impact can an evaluation performed accompanied by ultrasound have compared with a simple objective examination?

Response: Thank you for your suggestions. Ultrasound use in the prehospital setting is an emerging frontier with increased interest and adoption of this technology. Actually, several studies reported the use of ultrasound in disaster and emergency events. Portable ultrasound was used in the triage of patients after the 2010 Haitian earthquake. The investigators noted that the results of their ultrasound examination influenced care in 70 % of cases. Ultrasound use following the Wenchuan earthquake disaster in 2008 reported sensitivity of 91.9 % and specificity of 96.6 % for the diagnosis of abdominal injuries. (lines 142-146).

Point 4: How does a portable ultrasound machine differ in terms of image quality from one present in the emergency department?

Response: We sincerely appreciate your valuable comments and have added a table(1)  describing the diffences between portable handheld device and a high-end ultrasound device.(lines 47-52).

Point 5: Are there studies that have shown that ultrasound use reduces mortality or other end points in the prehospital setting? If not, add it as a limitation.

Response: Yes, there are several studies reporting the use of ultrasound in disaster and emergency events. Portable ultrasound was used in the triage of patients after the 2010 Haitian earthquake. The investigators noted that the results of their ultrasound examination influenced care in 70 % of cases. Ultrasound use following the Wenchuan earthquake disaster in 2008 reported sensitivity of 91.9 % and specificity of 96.6 % for the diagnosis of abdominal injuries. (lines 142-146).

Point 6: How much would it cost to equip each operating unit with a portable ultrasound machine? If it is too costly, might the use of portable ultrasound be advisable only for the most severe scenarios?

Response: Thanks for your comments. As the equipment gets closer to true “mass production”, it also becomes more readily accessible and less expensive(lines 318-319). 

Round 3

Reviewer 3 Report

Comments and Suggestions for Authors

All right, you have clarified my doubts. Good job.